# The Effects of Chloride Flux on *Drosophila* Heart Rate

**DOI:** 10.3390/mps2030073

**Published:** 2019-08-22

**Authors:** Catherine E. Stanley, Alex S. Mauss, Alexander Borst, Robin L. Cooper

**Affiliations:** 1Department of Biology, Center for Muscle Biology. University of Kentucky, Lexington, KY 40506-0225, USA; 2Max Planck Institute of Neurobiology, 82152 Martinsried, Germany

**Keywords:** optogenetics, *Drosophila*, heart, chloride channel

## Abstract

Approaches are sought after to regulate ionotropic and chronotropic properties of the mammalian heart. Electrodes are commonly used for rapidly exciting cardiac tissue and resetting abnormal pacing. With the advent of optogenetics and the use of tissue-specific expression of light-activated channels, cardiac cells cannot only be excited but also inhibited with ion-selective conductance. As a proof of concept for the ability to slow down cardiac pacing, anion-conducting channelrhodopsins (GtACR1/2) and the anion pump halorhodopsin (eNpHR) were expressed in hearts of larval *Drosophila* and activated by light. Unlike body wall muscles in most animals, the equilibrium potential for Cl^−^ is more positive as compared to the resting membrane potential in larval *Drosophila*. As a consequence, upon activating the two forms of GtACR1 and 2 with low light intensity the heart rate increased, likely due to depolarization and opening of voltage-gated Ca^2+^ channels. However, with very intense light activation the heart rate ceases, which may be due to Cl^–^ shunting to the reversal potential for chloride. Activating eNpHR hyperpolarizes body wall and cardiac muscle in larval *Drosophila* and rapidly decreases heart rate. The decrease in heart rate is related to light intensity. Intense light activation of eNpHR stops the heart from beating, whereas lower intensities slowed the rate. Even with upregulation of the heart rate with serotonin, the pacing of the heart was slowed with light. Thus, regulation of the heart rate in *Drosophila* can be accomplished by activating anion-conducting channelrhodopsins using light. These approaches are demonstrated in a genetically amenable insect model.

## 1. Introduction

Cardiovascular dysfunction is a major health problem in modern societies [1]. Some cardiac disorders involve issues in the electrical pacing of the heart [2,3]. In such disorders, electrical events are either arrhythmic or exhibit uncontrolled hyperactivity or hypoactivity. Currently, extrinsic electrical regulation of the mammalian heart is accomplished with wires placed, as electrodes, in contact with the heart [4]. These electrodes provide stimulating defibrillation to the tissue to either pace or reset the intrinsic electrical activity in the cases of cellular pacemaker failure or in atrial or ventricular fibrillation and tachycardia. These electrical devices are sewn or placed on the heart, which in some cases can cause irritation and damage to cardiac tissue over time [5,6,7]. Likewise, pharmacological treatments for these issues focus on both chronotropic and ionotropic function of the heart to regulate blood pressure and heart rate (i.e., beta blockers, calcium channel blockers, sympathomimetics, and parasympathetic blockers) and are also known to have off-target side effects. Pharmacological differences among African Americans and Caucasians are known in controlling hypertension such as for thiazides, beta blockers, and renin–angiotensin system inhibitors [8]. As for elderly people, antiarrhythmics (such as dronedarone) are known to produce enhanced side effects [9]. Many people do well with such standard therapeutic procedures and these therapies have prolonged human life. However, ionotropic control of the heart is difficult to manage with implanted electrical wires. Furthermore, prolonged inhibition of electrical activity is not possible with electrical defibrillation of the tissue for acute cardiac surgery or experimental studies. These limitations can potentially be overcome by using optogenetics to regulate the activity of pacing cells in desired directions [10,11,12,13]. 

The optogenetic revolution has been mostly exploited to alter brain activity and neural circuits with the potential treatment of disorders such as epilepsy and Parkinson’s [14,15,16,17]. However, recent breakthroughs suggest that optically regulating cardiac function is now also possible by either stimulating or inhibiting electrical signals without direct contact of the heart and without systemic pharmacological agents [10,18]. To this end, various forms of light-activated channels and pumps sensitive to different wavelengths of light and conducting different types of ions provide a means to control the activity of excitable cells.

Depolarizing light-sensitive channels (i.e., channelrhodopsin) have been utilized in oocytes of *Xenopus laevis* and mammalian cells [19,20] and in rodent neurons [21]. Likewise, anion channels have been used in rodent neurons and cell lines [22,23,24,25] as well as in cardiac cells [11,12]. The chloride-conducting channelrhodopsins (iC1C2, GtACR1/2) and halorhodopsin (eNpHR, a chloride pump) have allowed neural pathways to be specifically targeted to block neuronal activity [15,26,27] and show promise in a variety of cell types [11,12,25,28,29,30] including cardiac tissue. The cryptophyte algae species *Guillardia theta* expresses light-gated chloride channels (i.e., GtACR1 and GtACR2) [24], which can be used to create transgenic *Drosophila* lines that express these channels in specific cell types [25,31]. As the excitation wavelength is different for GtACR1 (515 nm) and GtACR2 (470 nm), differences in activation between these channels as well as selective expression of either one can be utilized to modulate cardiac function. Both forms have been shown to work within a fraction of second [11] and are reversible in silencing neurons in intact behaving flies [25,31]. These chloride channels can shunt the membrane potential toward the Cl^−^ ion reversal potential (*E_Cl_^−^_rev_*) [11,12]. Another light-sensitive protein, eNpHR, also transports chloride across the membrane but is different in that it is a pump rather than a channel [32]. Thus, Cl^−^ ions are transported into the cell even against the electrochemical gradient, which will induce hyperpolarization irrespective of the Cl^−^ equilibrium potential. 

*Drosophila* serve as a model for addressing many disease states, including cardiac dysfunction in humans [33,34,35,36,37,38,39]. Physiological parameters such as cardiac output, rate, and duration of systole and diastole in *Drosophila* are measurable [40]. The larval *Drosophila* heart is myogenic without direct neural connections, which simplifies investigations. Additionally, one can examine alterations in heart rate with in situ larval preparations bathed in saline in which modulators or ions within the hemolymph are removed or added [41,42,43]. The vast collection of genetically engineered lines for controlling cellular activity and manipulating gene expression make *Drosophila* a powerful model system [44,45]. The use of optogenetics in *Drosophila* has already been optimized for altering specific behaviors by driving channelrhodopsins in identifiable sets of neurons [25,46].

In addition, it has recently been demonstrated that heart rate can be increased in the *Drosophila* larva using the light-sensitive channelrhodopsin ChR2.XXL, a hypersensitive variant of ChR2 [47,48]. Here, we examine the possibility of decreasing chronotropic function with optogenetics. We investigated the ability to slow heart rate by activating the Cl^−^ pump and Cl^−^ channels in cardiac tissue in *Drosophila*. We approached this with an analysis of heart rate in intact and in situ preparations as well with measurements of the membrane potential in cardiac tissue. We found a similar effect among the optogenetic chloride effectors in that the heart rate could be effectively reduced with each of them. However, the underlying physiological changes were diametrically opposed: while the pump hyperpolarized heart muscles, activation of the channels led to depolarization. This can be explained by a chloride equilibrium potential above the resting membrane potential in heart muscles, in contrast to neurons. It is known that the *E_Cl_^−^_rev_* for body wall muscle of *Drosophila* larvae is more depolarized than the resting membrane potential [49]. Depolarization or hyperpolarization within cells by activating Cl^−^ channels is known to occur and can even make cells more excitable by lowering the threshold for producing action potentials [28,49,50,51,52,53,54]. 

Recently, the anion channels have been expressed in isolated cardiomyocytes of rabbits and in intact zebrafish hearts to alter the pacing of the isolated cells and intact hearts [12]. The current work demonstrates the potential of optogenetic chloride effectors in modulating heart rate. It also highlights the importance of considering the physiological consequences of applying optogenetic tools. Such procedures to diminish the heart rate may be possible to implement in intact mammals as well as humans in the future. 

## 2. Methods

### 2.1. Drosophila Lines

The filial 1 (F1) generations were obtained by crossing females of the recently created GtACR1^attP40^, GtACR1^VK00005^, GtACR2^attP40^ and GtACR2^VK00005^ lines [25] and the halorhodopsin line, w*; P{UAS-eNpHR-YFP} (BDSC# 41752), with male, non-stubble 24B-Gal4 (III) (BDSC stock # 1767). The GtACR1^attP40^, GtACR1^VK00005^, GtACR2^attP40^, and GtACR2^VK00005^ lines [25] and the halorhodopsin line, w*; P{UAS-eNpHR-YFP} without crossing to 24B-Gal4 (III) were also examined for the effect of light on the muscle as controls.

### 2.2. Preparation of Fly Food Supplemented with ATR

All-trans-retinal (ATR; Sigma-Aldrich, St. Louis, MO, USA) was diluted in 50 mL of standard fly food to a final concentration of 200 µM and protected from light with aluminum foil. A lower concentration can be used for GtACR1 and GtACR2 but activation of eNpHR requires a higher concentration, so a consistent 200 µM concentration was used throughout.

Retinal is used as a cofactor for the channel rhodopsin which increases its sensitivity to light and increases single channel conductance [55]. ATR (500 mg) was dissolved in 17.6 mL absolute ethanol to make 100 mM stock solutions. Stock solution was transferred to small tubes, wrapped with aluminum foil, and kept in a −20 °C freezer. The ATR was kept away from light, since it is light-sensitive. Fly food was dissolved in the microwave and was left to cool, then ATR was mixed with food or absolute ethanol in place of ATR as a control. Both foods were stored in the refrigerator for 3 days with a cotton plug to allow the ETOH to evaporate.

### 2.3. LEDs

The high intensity LEDs used in this study were blue light (470 nm wavelength, LEDsupply, LXML-PB01-0040, 500 mA), yellow-lime light (567.5 nm wavelength, LEDsupply, LXML-PM02-0000, 500 mA), or green light (530 nm wavelength, LED supply, 07007-PM000-L, 500 mA). 530 nm green light was used for the GtACR1 lines, 470 nm blue light was used for the GtACR2 lines, and 567.5 nm yellow-lime light was used for the eNpHR lines based on which wavelengths resulted in maximum activation of the light-sensitive proteins, determined from literature [24,25,56].

The photon flux (number of photons per second per unit area) was obtained with a LI-COR (model Li-1000 data Logger, LDL3774; LI-COR from Lincoln, NE, USA) which measured µMol s^−1^ m^−2^ µA^−1^. In addition, the full spectrum of the lights was measured with a Jazz (Ocean Optics Inc., Largo, FL, USA) to obtain a total W/m^2^ from 340 to 800 nm spectrum for each light source. 

### 2.4. Intact and Dissected Larvae

The intact larvae were restrained to one location by using double sided sticky tape on a glass slide and placing the ventral side of the larva on the tape [41]. The heart rates were counted by direct visual observation for 15 s before and 15 s during light exposure. The larvae were removed by placing water on the double-sided sticky tape (see video of procedure for details, [41]). High intensity light from the LEDs was focused on the specimen through a 10× ocular objective while the heart rate (HR) was counted [57]. The light was moved to a distance which produced a focused beam directly on the heart tube [57]. The light intensities are reported and the same procedure for the calibration device as for the larvae was used when determining the light intensities.

The general larval dissection technique to expose the larval heart tubes has been previously reported [41]. In brief, the larvae were dissected ventrally and pinned on four corners to expose the heart tube (Figure 1).

The visceral organs were removed, keeping the heart tube intact. This dissection technique was previously used to directly assess light-sensitive ion channels and pumps as well as pharmacological agents on the heart of Drosophila larvae [58,59,60,61,62,63,64]. Pinning of the animal on its back after dissection was used to directly apply the LED lights on the caudal aspect of the heart and to allow for the exchange of the bathing saline to one containing serotonin in later steps of the procedure. The dissection time was roughly 3–6 min, and the preparation was allowed to relax while bathed in saline for 3–5 min after dissection. The heart rate was monitored and recorded after the initial dissection for 15 s and during exposure to the appropriate light for 15 s. A modified HL3 saline was used to maintain the in situ hearts and body wall muscles (NaCl *70* mM, KCl *5* mM, MgCl_2_·6H_2_O 20 mM, NaHCO_3_ 10 mM, Trehalose *5* mM, sucrose 115 mM, BES 25 mM, and CaCl_2_·2H_2_O 1 mM, pH 7.1) [43]. For recording membrane potentials in cardiomyocytes, the saline mentioned above was used but without adding CaCl_2_·2H_2_O. Serotonin (1 µM) was used to increase the larval heart rate [42,65,66] in each experimental group so that it would be easier to verify changes in heart rate as well as to determine if activation of the light-sensitive Cl^−^ channels or pump altered the rate even in the presence of compounds which modulate cardiac function (salts for the saline and serotonin were obtained from Sigma-Aldrich Corporation, St. Louis, MO, USA). These procedures were repeated until ten larvae from each line were shown to respond to the light. Data from larvae that did not respond to the light were recorded as well.

### 2.5. Measures of Membrane Potential in Cardiac and Body Wall Muscles

To monitor the transmembrane potentials of the myocytes, the caudal region of heart was impaled with a sharp intracellular electrode (30 to 40 MΩ resistance) filled with 3 M CH₃COOK. An Axoclamp 2B (Molecular Devices, Sunnyvale, CA, USA) amplifier and 1 X LU head stage was used. Detailed procedures are shown in video format [40]. The saline was then exchanged to low Ca^2+^ saline. After the heart stopped beating (about 2 min) and after monitoring the resting membrane potential for a few minutes, the light at the appropriate wavelength for the lines was applied to the preparation for 10 s and the membrane potential was monitored. Each preparation consisted of two trials of light, separated by 3 min.

The effect on the membrane potential by activating the light-sensitive proteins was also examined in body wall muscle m6. Only the GtACR1 ^VK00005^ × 24B and GtACR2 ^VK00005^ × 24B lines for the anionic channels were examined. The body wall muscle m6 of the eNpHR × 24B line was also examined. Controls were the GtACR1^VK00005^, GtACR2^VK00005^ and eNpHR parental lines under the same conditions. 

Amplitudes of the spontaneous quantal events were measured for the eNpHR × 24B line from 50 quantal events immediately prior to the light pulse and all of the single quantal events during the light pulse. The amplitudes were measured as the voltage difference from the base to the peak of the events. Only quantal events with a distinct rapid rise were measured and care was taken not to measure events which were additive from superimposed quantal events as seen by deflections in the traces. 

### 2.6. Statistical Analysis

Some data are expressed as raw values. A Sign pairwise test was used to analyze changes in heart rate after changing bath conditions or with exposure to light. Since some data sets are not normally distributed (a number of zeroes in some groups) the non-parametric Sign test was used. When appropriate, paired and unpaired t-tests were used. A significant difference is considered *p* < 0.05. Different symbols were used in the graphs to isolate individual preparations from each other.

## 3. Results

### 3.1. Heart Rates within Intact Larvae 

The heart rates in each line were compared before and during light exposure with intact larvae held in position with double sided sticky tape. The two 3rd chromosomal lines (GtACR1^VK00005^ and GtACR2^VK00005^) were robust in stopping the heart very rapidly with 530 nm green (166.5 W/m^2^ or 930 µMol s^−1^ m^−2^ µA^−1^) and 470 nm blue (101.3 W/m^2^ or 1052 µMol s^−1^ m^−2^ µA^−1^) light, respectively (*n* = 10; *p* < 0.05; Figure 2A1,B1). Likewise, the 2nd chromosomal lines (GtACR1^attP40^ and GtACR2^attP40^), with the same light exposures performed on the same day and time period, showed similar responses (*n* = 10; *p* < 0.05; Figure 2C1,D1). The parental lines, which served as controls, were not sensitive to light exposure (Figure 2). The hearts of larvae in the GtACR1^attP40^ line started to quiver (fibrillation-like behavior) when exposed to the associated lights, and this behavior was observed in two of the 14 larvae from the GtACR2^attP40^ line as well. The eNpHR line was consistent in stopping the heart contractions upon 567.5 nm yellow (623.7 W/m^2^ or 2109 µMol s^−1^ m^−2^ µA^−1^) light exposure (*n* = 10; *p* < 0.05; Figure 2E). The expression levels of GtACR inserted on attP40 and VK00005 were not compared. The attP40 and VK00005 larvae did not differ in the trend of responses in each recording. The parental lines of GtACR1^VK00005^, GtACR2^VK00005^, GtACR1^attP40^, and GtACR2^attP40^ were also examined with the appropriate LED lighting. The heart rates in the parental lines did not show any differences in heart rates when exposed to the LEDs (Figure 2A2,B2,C2,D2,E2). The supplemental videos (Appendix A) capture a representative heart tube ceasing to beat with blue light exposure in a GtACR2^VK00005^ larva and yellow light exposure for an eNpHR-expressing larva).

### 3.2. Heart Rates within Dissected Larvae

The dissected preparations eliminate the potential influence of biogenic amines and peptides in the hemolymph which affect heart rate. However, perhaps because of this, the rates tend to be slower in dissected preparations as compared to measurements within intact larvae. This has been shown previously [62]. Variation in the heart rates among freshly dissected preparations was expected as this has been observed in previous studies. To aid in demonstrating that the heart rate was altered by light exposure, the heart rates were sped up by exposure to serotonin (5-HT) at 1 µM. It is established that 5-HT will rapidly increase the heart rate and maintain a high rate for several minutes in CS larvae [42,63,64]. With the accelerated rate, the effect of activating the light-sensitive channels was more readily discerned. All the larvae for GtACR1^VK00005^ and GtACR2^VK00005^ had an increased heart rate with 5-HT exposure and all hearts ceased to beat with the associated full light exposure (Figure 3; each *n* = 10; *p* < 0.05, Sign test). A video (Appendix A) captures a representative heart tube ceasing to beat with blue light exposure in a dissected GtACR2^VK00005^ larva. 

The larval hearts of the eNpHR line all stopped beating with 567.5 nm yellow light exposure in saline without and with 5-HT (1 µM) (Figure 3E, *n* = 10; *p* < 0.05, Sign test). A video supplement (Appendix A) illustrates a representative heart tube ceasing to beat with yellow light exposure in a dissected eNpHR -expressing larva. All the preparations also exhibited an increased heart rate with exposure to 5-HT. 

The light intensities used on the dissected preparations were the same as for the intact larvae; however, the exposed hearts in saline might receive a slightly higher level due to the absence of the transparent cuticle. It is not known how much, if any, the thin cuticle in larva may attenuate the blue, green and yellow lights.

One interesting observation occurred in only the GtACR1 and GtACR2 lines. As the focused light beam of LED approached the heart tube from the periphery, the heart rate would rapidly increase; when the beam was fully focused on the heart, however, the full contraction would stop and in some cases the heart tube would quiver. This appeared as a fibrillation of very small contractions, but they were not rhythmic along the heart. This is best illustrated in a supplemental video (Appendix A) for GtACR1^VK00005^ × 24B which was treated with 5-HT (1 µM) to increase the heart rate and exposed to a gradually increasing intense green light (166.5 W/m^2^). Additionally, a representative GtACR2^VK00005^ × 24B larva is shown for illustrating the effect of dim white light changing to more intense white light (0.425 W/m^2^) and dim blue to high intensity blue light (101.3 W/m^2^) (Appendix A) in which this effect also occurs.

Since an increase in the heart rate was noticed with peripheral green or blue light for the GtACR1^VK00005^ and GtACR2^VK00005^-expressing lines, respectively, a systematic study was performed with very dim, medium, and high intensity light (Figure 4). The varied light intensity was also performed for the eNpHR line even though the rate did not show an increase when the high intensity light was approaching the specimens from the periphery. The intensity of the blue light was increased from dim intensity (2.1 W/m^2^) to 6.02 W/m^2^, and both of these levels increased heart rate. With high intensity light (101.3 W/m^2^) the heart stopped (Figure 4A, *p* < 0.05, two tailed T-test). The green light was also tested with a dim (2.1 W/m^2^) to medium (7.68 W/m^2^) and then to high intensity (166.49 W/m^2^) and a similar trend occurred, with the lower intensities increasing heart rate and high intensity causing the heart to cease beating (Figure 4B, *p* < 0.05, two tailed T-test). A low intensity of yellow light (2.98 W/m^2^) did not have an impact on the eNpHR-expressing larva. However, the medium intensity (26.45 W/m^2^) and high intensity (623.67 W/m^2^) light decreased the heart rate (Figure 4C, *p* < 0.05).

### 3.3. Membrane Potential in Cardiac and Body Wall Muscles

Membrane potentials were obtained in muscle m6 of the body wall muscles in response to light exposure in the same lines as those used for measuring heart rates. The m6 body wall muscle is one of the wider and more readily accessible body wall muscles on the ventral side of larvae [67].

#### 3.3.1. Body Wall Muscles

The body wall muscles were not voltage clamped in order to mimic the natural variation among larvae and to relate to the potential conditions in intact larvae for future experiments on body wall muscles during development utilizing these genetic lines. Due to the expected differences in driving gradients for Cl^−^ ions at different resting membrane potentials, the potentials are listed for the maximum response during a 10 s light exposure. The change in the membrane potential with the application of the associated light for the GtACR1^VK00005^ and GtACR2^VK00005^ lines revealed, in the majority of cases, a depolarization in the body wall muscles as expected given the chloride reversal potential is less negative than the resting membrane potential [49]. 

A representative recording of the membrane potential in the body wall muscle of a larva from the GtACR2^VK00005^ line indicates the depolarization is rapid and sustained during the 10 s exposure (blue light at 8.8 W/m^2^). The return to the initial resting membrane potential is also rapid when removing the light (Figure 5A). The rate of change in the membrane potentials do vary slightly among preparations. After three minutes of no light exposure following the initial application of light, secondary light exposures produced similar responses. 

The same trend was observed in six of the seven preparations examined for the GtACR2^VK00005^ line (*p* < 0.05, Sign test; Green at 1.5 W/m^2^). One preparation demonstrated hyperpolarization with the two blue light pulses (Figure 5B). Additionally, the quantal responses decreased in amplitude during the pulses in spite of the membrane hyperpolarization which would have produced a greater driving gradient for Na^+^ through the glutamatergic receptors.

One preparation of the GtACR1^VK00005^ line initially showed a hyperpolarization, but after three minutes of dark adapting the response presented with an overall depolarization. In 11 preparations only one exhibited this unusual response. Thus, 10 out of 11 preparations in the GtACR1^VK00005^ line depolarized, as similarly observed for the GtACR2^VK00005^ line, with the two light pulses provided to each preparation (*p* < 0.05, Sign test). The change in the membrane potentials in each preparation for each larva in the GtACR1^VK00005^ and GtACR2^VK00005^ lines is shown (Figure 5C). The changes vary due to the differences in the initial resting membrane potentials. As mentioned above, the membrane was not voltage clamped to illustrate the natural variation possible in intact larvae. The focus of this procedure was to test the hypothesis that activating the GtACR1 and GtACR2 channels with light results in a depolarization of the body wall muscles.

The eNpHR line has been investigated in earlier studies and, as shown before, yellow light exposure (8.0 W/m^2^) results in hyperpolarization of the body wall muscles (Figure 6A). This was the case in 7 of 7 preparations examined in the current experiment (P < 0.05, Sign test). Additionally, note the spontaneous quantal responses of vesicle fusion events in the absence and presence of light. The quantal responses are readily observed during the hyperpolarization with the activation of the eNpHR pump. As shown for the GtACR1^VK00005^ and GtACR2^VK00005^ lines, the change in the membrane potentials in each preparation for eNpHR varies due to the differences in the initial resting membrane potentials (Figure 6B).

The amplitudes of the spontaneous quantal events during the depolarizing phases induced by light for the GtACR2^VK00005^ × 24B and GtACR1^VK00005^ × 24B lines are so small that they are not feasibly able to be measured, but it is evident that they are much reduced. An example is illustrated for a preparation of the GtACR2^VK00005^ × 24B line (Figure 7A1,A2,A3). The amplitudes of the spontaneous quantal events for eNpHR were also compared before and during the light pulse (Figure 7B1,B2,B3). The mean amplitude of the spontaneous quantal events was significantly larger during the hyperpolarization for five of the seven preparations examined (Paired T-test, *p* < 0.5 for each of the 5 preparations). A percent change in the amplitudes of the spontaneous quantal events for each of the seven preparations is depicted in Figure 7C and the overall mean percent change for the seven preparations was significantly increased (last bar in Figure 7C; Two tailed paired T-test, *p* = 0.014).

#### 3.3.2. Cardiac Muscle

Measuring the membrane potential in the cardiac muscle is challenging due to the very small size of the muscle fibers and the heart being suspended in the dissected larvae. In the time it would take to dissect and then place an intracellular electrode into cardiac muscle, the heart would generally stop beating. This appeared to be due to the higher intensity of white light exposure needed for the dissection and electrode placement compared to the level used to count the heart beats. The most caudal end of the heart tube does not move as much as the rest of the heart tube, and this is the region of the main pacemaker cells [41,62]. This reduced movement is due to the dissection pins placed in the caudal regions of the cuticle, on the side of the larvae off center from the heart tube, to hold the preparation in place. This region was the focus for obtaining intracellular recordings. However, these in situ hearts exposed to the saline containing 1 mM Ca^2+^ would beat when dark-adapted within a 2 min period after impaling the cells. In the lowered calcium saline, the hearts would generally stop beating in a few minutes. The movements made it very difficult to obtain stable recordings to assess the effect of light on the membrane potential. Since the heart rates in GtACR1^VK00005^ and GtACR2^VK00005^ lines were generally low, it was possible to impale cells for short periods to obtain a measurement of the membrane potentials and the sensitivity of the preparations to light. Rhythmic activity of the hearts was first established for GtACR1^VK00005^ and GtACR2^VK00005^ larvae after dissection. They were then exposed to light to determine its effect on rhythm and membrane potential. A preparation in which the beats caused a large deflection in the recording is shown in Figure 8. Upon blue light exposure the cell depolarized (from −27 to −20.8 mV in this example) and the strong beating ceased. However, note the small rapid fluctuation in the potential during the light exposure. This small amplitude, rapid fluctuation was also observed in other preparations and is likely related to the quivering movements as observed in the intact and dissected preparations (see Appendix A). In both the GtACR1^VK00005^ and GtACR2^VK00005^ lines, six out of six preparations for each line showed depolarizations with the associated light exposures (*p* < 0.05, Sign test). The resting membrane potentials in the heart cells showed variation and varying degrees of changes with light exposure. The variation in resting membrane potentials was also shown in earlier reports for CS larvae [41,62]. The differences in the potentials before and during light exposure are shown for each recording in both lines (Figure 8B1,B2).

Since the hearts of larvae in the eNpHR line beat rapidly in normal saline at 1 mM Ca^2+^, as compared to the GtACR1^VK00005^ and GtACR2^VK00005^ lines, the rate was reduced by exchanging the saline to the saline with no added calcium [65]. The hearts would not beat after dark adaptation for 2 min. In some cases, right after changing the saline to the reduced Ca^2+^ the heart would beat with reduced force and the electrical activity could still be recorded as shown in Figure 9A1. The yellow light exposure would cause the beats to cease and begin beating again upon removal of the light. Very slight hyperpolarization, if any, was observed within the 10 s of yellow light exposure. Other preparations with similar membrane potentials did demonstrate a hyperpolarization during the light exposure (Figure 9A2). The differences in membrane potential for each preparation are shown in Figure 9B. No systematic difference in the initial membrane potentials is noted among GtACR1^VK00005^, GtACR2^VK00005^, and eNpHR lines.

Examination of possible photoelectric effects with the electrodes and ground wire by the LED light sources revealed no changes in electrical potentials. The tip of the intracellular electrode was removed from the cells being recorded within 0.2 to 0.4 mm and the same light sources were used in examining if there was a photoelectric effect.

## 4. Discussion

In this study, we demonstrate a proof of concept to control heart rate in *Drosophila* by manipulating the flow of chloride ions with light. Contractions could be abolished by light-gated opening of the chloride channels GtACR1 and GtACR2 despite the equilibrium potential for chloride being less negative than the resting membrane potential of the cardiac cells, resulting in depolarization. The shunting of the membrane potential, when the channels are open, prevented further depolarization and thus eliminated the pacemaker-driven rhythms and contractions rapidly and steadily over 10 s periods. A similar mechanism of preventing depolarization-induced contractions would likely occur in mammals with the opening of the GtACR1 and GtACR2 chloride channels. Excitation of GtACR1 expressed in isolated cardiomyocytes of rabbits and in intact zebrafish hearts resulted in depolarization of the cells and cessation of pacing activity [12]. As Kopton et al. [12] suggested, the depolarized state is likely responsible for blocking the re-excitation and conduction which reduces the pacing activity. Similar observations made in this study with GtACR1 and GtACR2 chloride channels expressed in larval *Drosophila* hearts support this notion.

Activating the eNpHR chloride pump also rapidly reduced the heart rate and, similar to the chloride channels, the rate would return in the absence of the light. The effects on the membrane potentials for the cardiac cells in the eNpHR -expressing line paralleled the hyperpolarized responses obtained for the body wall muscles; however, the responses were more robust in the body wall muscles. The quivering or slight fluttering of the cells when the light pulses were given is interesting to note. The slight fluttering in the intact larvae is not as easily seen as in the dissected preparations, as the trachea is not pulled taut and the salivary glands and intestinal tissues are not stationary. In the dissected preparations a clear view of the heart tube and the edges of the tube are present. As seen in a supplemental video (Appendix A) of a dissected larva, the edges of the heart tube flutter very slightly by fractions of a millimeter, while the heart has ceased full contractions. The fluttering does not appear to be synchronized along the heart tube but rather randomly occurring at a high rate. The quivering was not observed when activating the eNpHR pump; thus, the phenomenon may be related to depolarization of the membrane. It would be interesting to determine if Ca^2+^ is fluxing in the cells during this time resulting in the localized miniature contractions. We are currently investigating this possibility with Ca^2+^ indicators but conditions need to be worked out as to not activate the light-sensitive proteins while imaging the localized changes in Ca^2+^ flux. 

The varying amounts of membrane depolarization for the GtACR1^VK00005^ or GtACR2^VK00005^ and hyperpolarization for the eNpHR larvae in the cardiac and skeletal muscles within a given line may well be due to altered expression levels of the proteins. This is challenging to address in the small amount of cardiac tissue but possibly can be done with pooled body wall muscles within a larva to compare to other larva within a line. It did not appear to be possible to voltage clamp the cardiac cells in the conditions used in this study and, even with body wall muscles of similar resting induced by opening the light sensitive GTACR2-chloride channels. This is likely due to the shunting of the membrane potential close to the equilibrium potential for chloride. When the Na^+^-induced depolarization current occurs from the opening of the ionotropic glutamate receptors during the quantal response, the potential would be driven to the equilibrium potential for chloride, preventing any depolarization. In addition, the quantal responses would be smaller due to a decreased driving gradient for Na^+^ influx during the light-induced depolarization. However, with the hyperpolarization the quantal responses showed an increase in peak amplitude. The membrane potentials for the light-induced responses still varied with the same consistent light pulses. The small spontaneous quantal responses of vesicle fusion events on the body wall muscles in the absence of light are not present during these light induced depolarizations.

The optogenetic effects may depend strongly on the cell type. The Cl^−^ pump mediates hyperpolarization in body wall muscle and to a lesser extent in cardiac muscle; conversely, activating the Cl^−^ channels mediates hyperpolarization in neurons [25,54] but depolarization in larval *Drosophila* body wall muscles. This complements previous accounts in vertebrates of developmental differences in the reversal potential (e.g., GABA receptors being first excitatory then inhibitory [68]). However, the *E_Cl_*^−^*_rev_* in skeletal muscle of many animals is more negative than the resting membrane potential. It is also known that crustacean and rodent skeletal muscles have a more negative equilibrium potential for Cl^–^ than the resting membrane potential [69,70,71]. Early work with Purkinje fibers from sheep or dog hearts [72,73,74] demonstrated that the Cl^–^ conductance, compared to the total membrane conductance, is much smaller in cardiac muscle than it is in skeletal muscle. This supports the differences we have noted in the *Drosophila* body wall muscle and cardiac muscle in regards to the membrane potential changes when activating the light sensitive Cl^–^ channels. 

Even different regions of neurons have varied equilibrium potentials for ions (e.g., soma vs. axon terminal, [53]). The use of expressing non-native channels in *Drosophila* body wall muscles has led to the discovery that the equilibrium potential for Cl^−^ is less negative than the resting membrane potential [49]. The results herein support this and indicate that the equilibrium potential for Cl^–^ is also less negative in the cardiac muscles. Since attP40 and VK00005 larvae behaved in a similar fashion with light exposures, the expression levels of GtACR inserted on attP40 and VK00005 were not necessary to compare. The PhiC31 integrase genomic landing sites attP40 and VK00005 on two different chromosomes were chosen to support reliable expression and maximal flexibility for crossing with other transgenes/drivers [75]. Resulting expression levels between the two sites were not compared in a quantitative way. Subtle differences might exist, which would however not affect conclusions from experimental results. In principal, any insertion site could have been chosen which supports reliable expression.

Even if the equilibrium potential for the Cl^−^ ion has a more depolarized value than the resting membrane potential, as is the case for the body wall muscle of the larval *Drosophila*, hyperpolarization is induced with an anionic^-^ pump since Cl^−^ ions are pumped into the cell [49,52]. Consequently, activation of GtACR1 and GtACR2 in the body wall muscles of larval *Drosophila* promotes depolarization to the Cl^–^ equilibrium potential. Any further depolarization while the channels are open, however, would be dampened, which occurs for presynaptic inhibition at synapses as well as on crustacean skeletal muscle [76,77,78]. 

The heart rate of *Drosophila* is generally reduced in dissected larvae compared to intact larvae. This is likely due to the removal of modulators and a saline which still does not exactly mimic the ionic composition of the hemolymph. The modified HL3 saline [43] does maintain the heart rate for a much longer duration than the initially derived HL3 saline but still lacks hormones and modulators. Activation of neurons using optogenetics, which likely resulted in a rise in the concentration of serotonin, dopamine, or acetylcholine in the hemolymph, increased heart rate in intact larvae [58]. It is known that each of these cardioactive substances increases heart rate when applied in isolation to the exposed hearts [42,63,79,80,81] and even in hearts with very reduced rates due to cold exposure [81]. In *Drosophila* larvae expressing the channelrhodopsin protein ChR2.XXL in cardiac tissue, light was shown to increase heart rate even in larvae cold conditioned (at 10 °C) or acutely exposed to cold. Given that 5-HT increased the rates substantially and opening the chloride channels or activating the chloride pump with light could essentially stop the hearts from beating, this indicates that this optogenetic method is a very effective means of controlling the heart even under conditions which accelerated the rate by a second messenger [63,79].

Past pharmacological and RNA interference studies on larval *Drosophila* hearts demonstrated that 5-HT2ADro and 5-HT2BDro receptor subtypes are the mechanism in which 5-HT increases heart rate in the larvae. The mechanism by which this occurs is likely through G-protein coupled receptors (GPCR) since the 5-HT2 receptor is coupled to the Gαq protein in mammals. Gαq can activate phospholipase Cβ (PLC) and thus lead to the creation of second messengers, inositol triphosphate (IP3) and diacylglycerol (DAG). The DAG can activate protein kinase C (PKC) and IP3 targeting the endoplasmic reticulum (ER) to raise cytosolic Ca^2+^. The 5-HT was also shown not to be working through adenylyl cyclase in *Drosophila* larval hearts [79,80]. Thus, the ability to control Cl^−^ flux through the membrane is overriding these intracellular cascades in order to have the effect of increasing the rate. 

Obtaining consistent and repeatable responses over time from the optically sensitive channels is important for potential therapeutic treatments (see review [82]). The possibility of the expressed proteins being targeted in other membranes besides the surface of the cell has not been fully investigated with the variety of light-sensitive channels being expressed in non-native organisms. For example, it is known that nicotinic acetylcholine receptors (nAChRs) can occur on endoplasmic reticulum (ER)-derived microsomes [83], which suggests cell trafficking of proteins can result in ion channels being directed to other localizations besides the cellular plasma membrane and still maintain function [84]. The alteration of Cl^−^ ion levels within cells likely has consequences on other ionic fluxes. Increasing the Cl^−^ concentration may have an impact on cellular pH by driving the Cl^−^/HCO_3_^−^ exchanger. Other ion levels may also change if voltage-gated channels are opened during the membrane potential change induced by activating the light-sensitive channel. Such could be the case for the funny voltage-gated sodium channels which open upon hyperpolarization in mammalian heart tissue [85] or even the hyperpolarization-induced Cl^−^ channels in *Drosophila* skeletal muscle [49]. 

The acute and chronic level of expression over time may modify the sensitivity of the light-sensitive response. Thus, future studies on this matter would be of interest. Research on using optogenetics as gene therapy for potentially controlling atrial and ventricular tachycardia and in pacing mammalian hearts is occurring in mammalian models and shows exceptional therapeutic promise. Modifications of chronotropic and ionotropic control of the heart in model animals using an array of light-sensitive ion channel types and pumps to excite and inhibit cellular activity will be an active area of research for the next several years.

These results and others demonstrate that optogenetic expression of light-sensitive proteins can increase or decrease the heart rate in a variety of conditions, such as when the rate is enhanced with a modulator or depressed by cold temperature or in pathological conditions.

## Figures and Tables

**Figure 1 mps-02-00073-f001:**
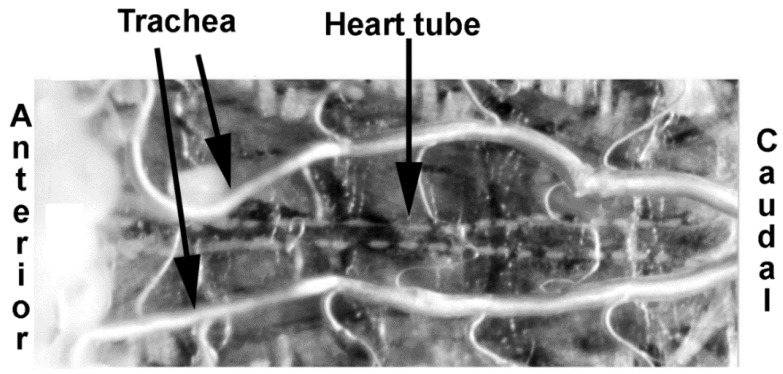
Ventral dissection representation of a third instar larva.

**Figure 2 mps-02-00073-f002:**
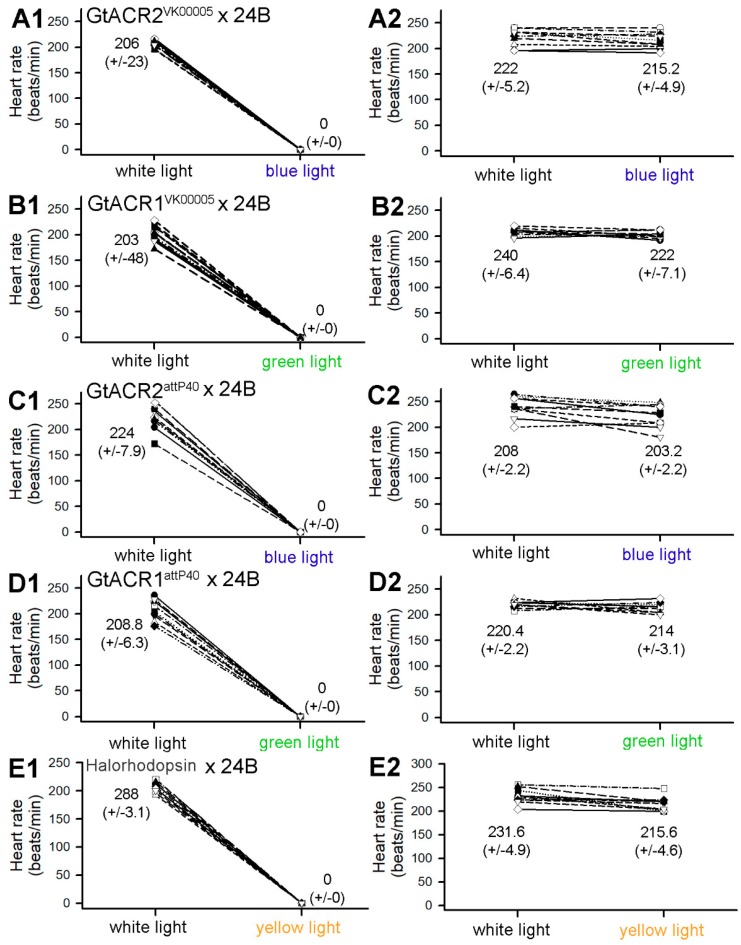
Heart rates of intact larvae from each line before and during exposure to the appropriate wavelength of light. Heart rates for all larvae in the GtACR2^VK00005^ × 24B line (**A1**), the GtACR1^VK00005^ × 24B line (**B1**), the GtACR2^ATTP40^ × 24B line (**C1**)**,** the GtACR1^ATTP40^ × 24B line (**D1**), and the Halorhodopsin (eNpHR) × 24B line (**E1**) decreased to zero after direct exposure to the light. The parental lines (**A2**, **B2**, **C2**, **D2**, **E2**) showed no difference to the light exposure. Green (166.5 W/m^2^), blue (101.3 W/m^2^) or yellow (623.7 W/m^2^) light was used. Mean (+/-SEM) values are presented next to each distribution.

**Figure 3 mps-02-00073-f003:**
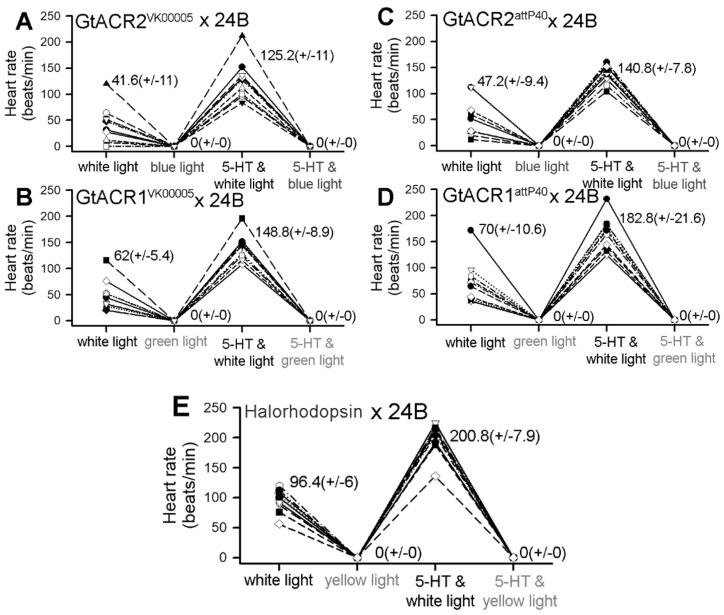
Heart rate of dissected larvae before and during exposure to the appropriate wavelength of light and the effects of serotonin (5-HT). Heart rates for all larvae in the GtACR2^VK00005^ × 24B line (**A**), the GtACR1^VK00005^ × 24B line (**B**), the GtACR2^ATTP40^ × 24B line (**C**), the GtACR1^ATTP40^ × 24B line (**D**), and the Halorhodopsin (eNpHR) × 24B line (**E**) decreased to zero after direct exposure to the light, increased after exposure to 5-HT, and decreased back to zero after direct exposure to the light in the presence of 5-HT. Green (166.5 W/m^2^), blue (101.3 W/m^2^) or yellow (623.7 W/m^2^) light was used. Mean (±SEM) values are presented next to each distribution. Each group is significantly different from the other groups except between the distributions in which a light is used without and with 5-HT. (*p* < 0.05, Friedman repeated measures ANOVA and a post hoc Bonferroni t-test).

**Figure 4 mps-02-00073-f004:**
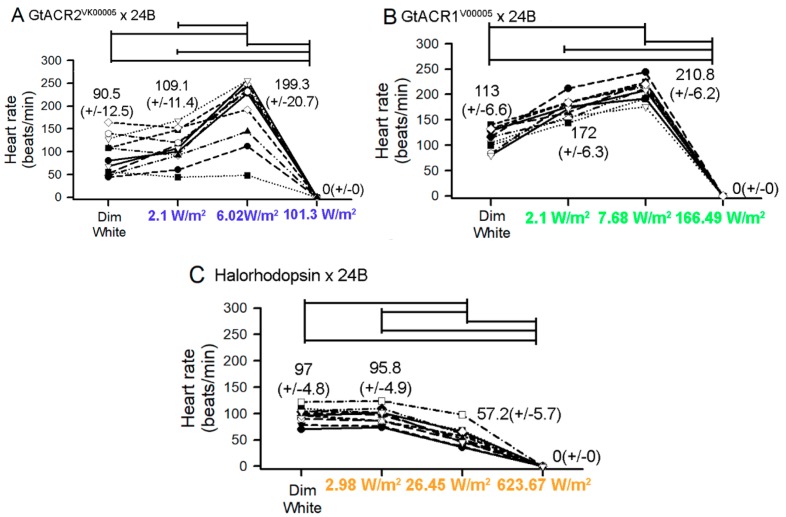
Heart rates of dissected larvae from GtACR2^VK0005^ × 24B, GtACR1^VK0005^ × 24B, and Halorhodopsin (eNpHR) × 24B lines before and during exposure to the appropriate wavelength of lights at varying intensities. Mean (+/-SEM) values are presented next to each distribution.

**Figure 5 mps-02-00073-f005:**
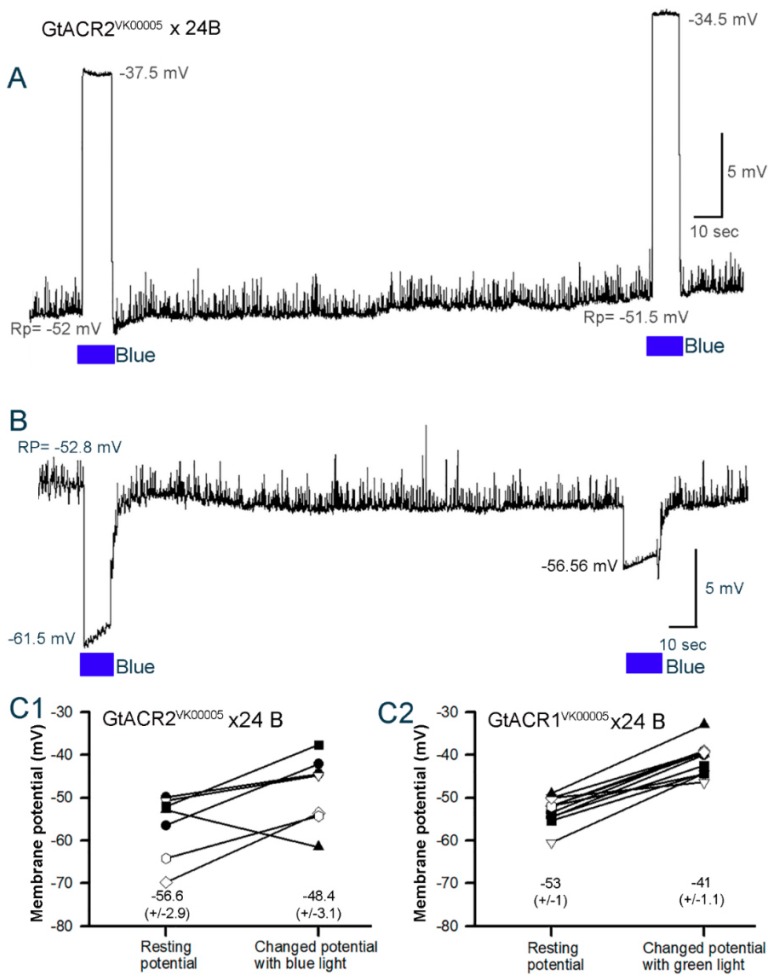
Membrane potential changes in body wall muscle m6 in 3^rd^ instar larva for GtACR2^VK00005^ and GtACR1^VK00005^ lines. A representative trace of membrane potential depolarization upon activation with blue light (**A**) and a rare occurrence of hyperpolarization (**B**). The light pulse is 10 s. Note the spontaneous quantal synaptic responses during dark conditioning between pulses and the lack of such occurrences during the light pulses. The change in membrane potentials from resting condition to peak response during the 10 s light pulse for GtACR2^VK00005^ (**C1**) and GtACR1^VK00005^ (**C2**) lines is shown for each preparation. Blue light at 8.8 W/m^2^ or green light at 1.5 W/m^2^ was used. Mean (±SEM) values are presented next to each distribution.

**Figure 6 mps-02-00073-f006:**
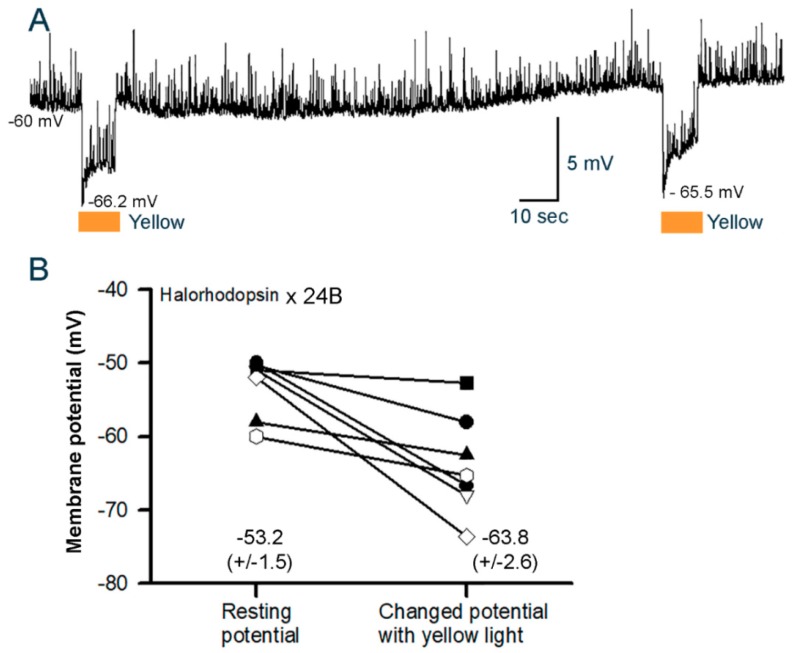
Membrane potential changes in body wall muscle m6 in 3rd instar larvae for the halorhodopsin (eNpHR)-expressing line. (**A**) The membrane potentials of larva expressing eNpHR hyperpolarize when exposed to yellow light. The light pulse is 10 s. Note the spontaneous quantal synaptic responses during dark condition between pulses and during the light pulses. (**B**) The changes in membrane potential from resting condition to peak response during the 10 s light pulse (8.0 W/m^2^ yellow light) for each preparation. Mean (± SEM) values are presented next to each distribution.

**Figure 7 mps-02-00073-f007:**
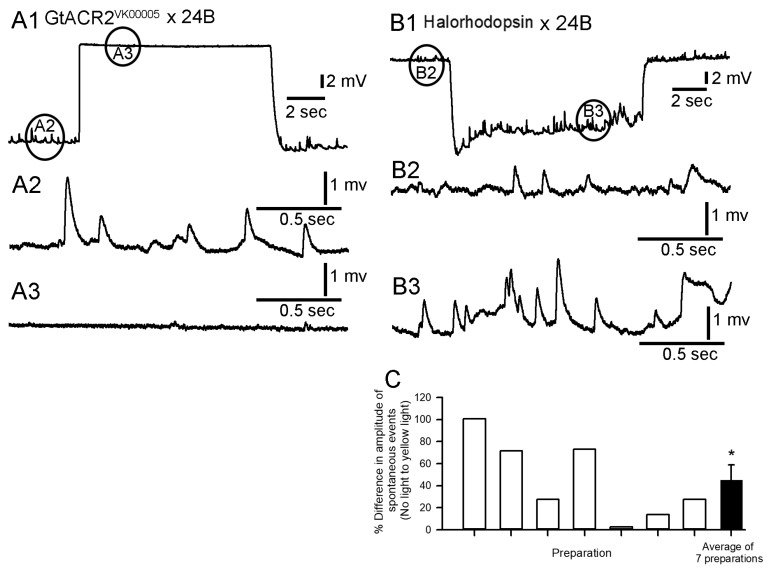
Amplitudes of spontaneous quantal events before and during light pulses. (**A1**) The spontaneous events are readily observed before the light pulse (**A2**) for a GtACR2^VK00005^ × 24B larva, but during the depolarization induced by blue light (**A3**) are greatly reduced. (**B1**) The halorhodopsin (eNpHR) × 24B-expressing line shows quantal synaptic events before (**B2**) and during (**B3**) the hyperpolarizing event induce by yellow light. (**C**) The percent difference in the mean amplitude of spontaneous quantal events before and during the hyperpolarizing light pulse for the seven halorhodopsin (eNpHR) × 24B-expressing preparations (Two tailed paired T-test, *p* = 0.014). The circles in A1 and B1 indicate where the graphs were enlarged for representative activity before (A2, B2) and during (A3, B3) the light pluses.

**Figure 8 mps-02-00073-f008:**
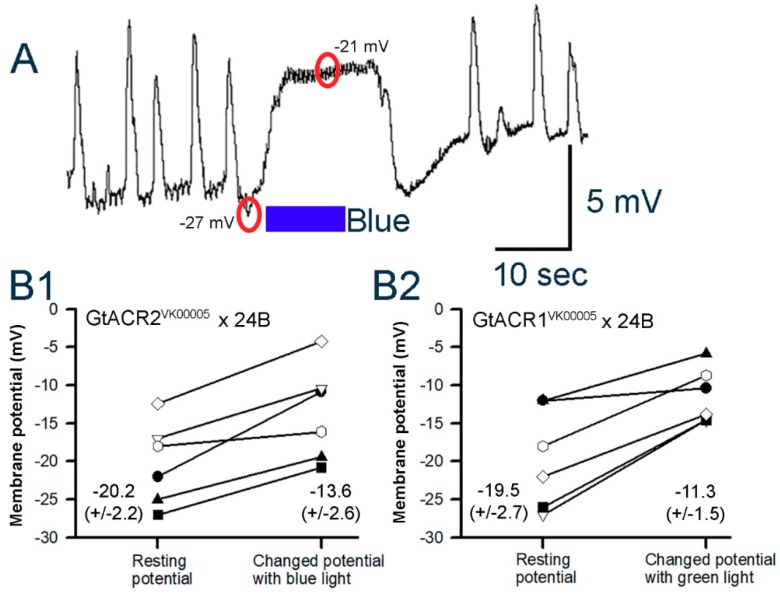
Membrane potential changes in heart muscle 3^rd^ instar larva for GtACR2^VK00005^ × 24B and GtACR1^VK00005^ × 24B lines. (**A**) The heart is rhythmically beating while recording with an intracellular electrode in a GtACR2^VK00005^-expressing larva. Upon the blue light pulse the membrane depolarizes and the beating ceases. Note the small fluctuation in the membrane potential during the light pulse and for a short while after the light is off. The change in membrane potentials of the heart from resting condition to peak response during the 10 s light pulse for GtACR2^VK00005^ (**B1**) and GtACR1^VK00005^ (**B2**) lines is shown for each preparation. The circles overlaying on the trace indicate where the membrane potential is measured. Mean (± SEM) values are presented next to each distribution.

**Figure 9 mps-02-00073-f009:**
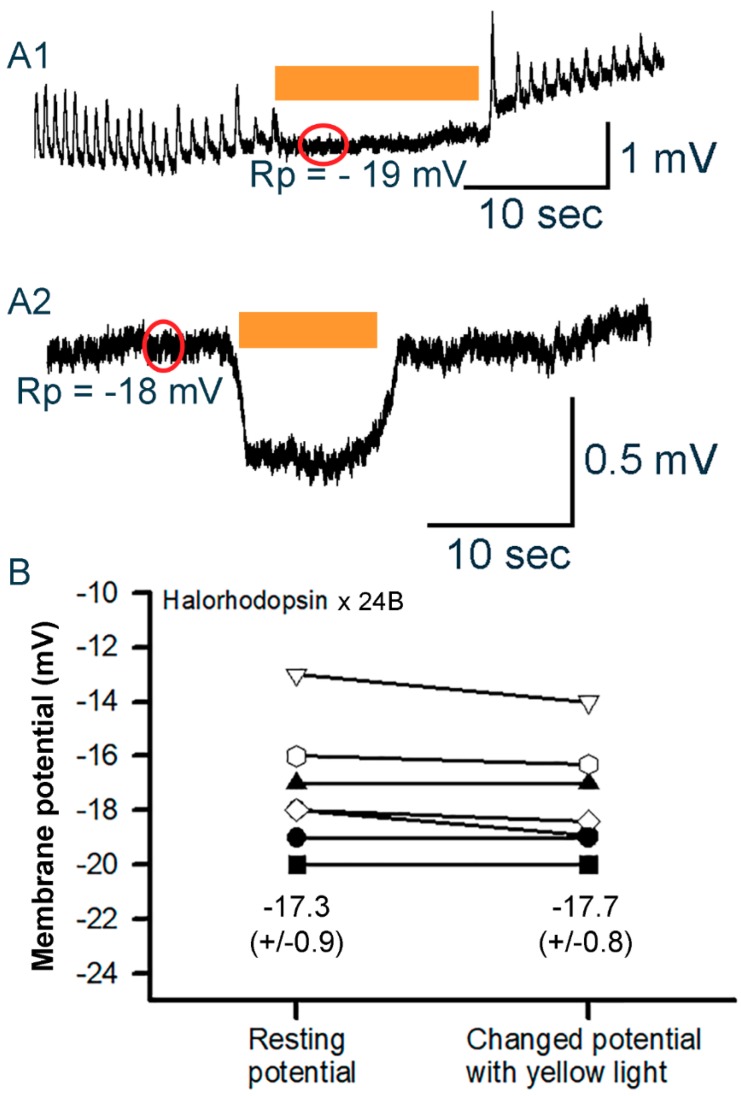
Membrane potential changes in heart muscle in 3rd instar larvae for the halorhodopsin (eNpHR)-expressing line. (**A1**) When halorhodopsin is activated by yellow light, the heart stopped beating without the membrane hyperpolarizing. (**A2**) This preparation shows a sustained hyperpolarization during the yellow light pulse. (**B**) The changes in membrane potential from resting condition to peak response hyperpolarizing potentials during the light pulse are shown for each preparation. The circles overlaying on the trace indicate where the membrane potential is measured. Mean (± SEM) values are presented next to each distribution.

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
