# Peer review of "The Effects of Chloride Flux on Drosophila Heart Rate"

_mps, 2019, doi:10.3390/mps2030073_

Round 1

Reviewer 1 Report

In this manuscript Stanley et al. present and characterized utility of anionic channelrhodopsin GtACRs, in reversibly slowing down heart rate in Drosophila. The authors utilized previously published neurogenetic tools like GtACR compared the magnitude of the effect induced with respect to another inhibition tool, eNpHR and monitored heart rate in both restrained and dissected larvae. In the manuscript, the authors present the proof of concept experiments examining the utility of the potent optogenetic silencing tools not only in freely moving, and dissected animals but also in animals in which heart rate was increased using 5-HT in the bath solution.  In general, I find the provided method interesting and useful. Specifically the electrophysiological comparison between GtACR and eNpHR and finding a mechanistic difference between the two light-gated channels and pump.

However, I feel that authors should be somehow more thorough in their writing, examination, and quantification of behavior. I believe the points raise below by me will be helpful for authors to improve this manuscript, which have the potential to get good citations.

Specific points:

Title

I felt as the manuscript has examined effects on heart rate, the stress point on heart rate will read better if authors like then the title could be “The effects of chloride flux on Drosophila heart rate

Introduction:

Although      the introduction is well written, it lacks crucial references which the      reader needed to understand the importance of work. For example, the very      first and second sentence (line 32-33), is a big statement, the authors      should substantiate it with references. There are many such instances.

In line 40-42, Authors tried to      highlight the the possible issues in using pharmacological agents in      controlling blood pressure and pacing, without naming even one issue or      side-effect off-target effect?

Methods:

Line 103-105, In the section “preparation of fly food supplemented with ATR” authors mentioned that they dissolved  ATR directly in the fly food, ATR is sparingly soluble in water, it is usually first dissolved in 90-100% ethanol, which is mixed in the medium for fly treatment.

Statistical Analysis:

In Line 157-1661, authors mentioned that data is expressed as either raw value or mean + SEM, in the paper, there is not even a single figure where data is presented as mean + SEM, in fact, all figure are presented as the raw value. In test Sign test has been mentioned, but its use is not mentioned in the statistical analysis section

Figures

As mentioned above, the authors plotted all graphs as raw data without providing mean and error. On top of that, in all figures, raw data is plotted with both open and filled features ( circle, square and triangles), without providing their details in legend.

In all figures, raw data of flies expressing GtACR1 or eNpHR in 24B-Gal4      expressing cells, with or without lights were plotted, no responder or driver control was used or shown. Authors did not mention why they selected 24B-Gal4, and not Tin-Gal4 or Hand-Gal4 which are used to label cardiac tubes in Drosophila.

Are the light intensities used were based on published data? If yes please cite. Overall light intensities used were in a higher range keeping transparency of larvae in view.

Authors provided videos as supplementary information, in line 178-1809, regarding video S2, authors mentioned that expression of eNpHR in 24B cells stopped heart rate in yellow light. However, in the video one can observe beating heart even in the presence of yellow light. Please re-check your videos.

Authors observed that while GtACR lines inserted on the 3rd chromosome were very potent compared to the lines inserted on the 2nd chromosome, however,      these results were not discussed in details like is the expression of      GtACR inserted on attP40 is weaker compared to the one on VK00005? This also suggests that authors should have used tagged GtACRs published earlier by other groups.

Providing videos is good for the readers, it will also be good to have a schematic or real picture of the prep used for counting the heart rate.

Similar to figure 1, figure 2-4 can be combined.

Line      229-230, authors mentioned some data in which they showed that some preparations were not the response to the light without pointing to the figure/s where the data is shown. Similarly, in line from  239-242 authors have not provided information about the figure or panel.

While for behavior authors show data of GtACR lines inserted on different chromosomes but for membrane potential measurement they chose only lines inserted on the 3rd chromosome, does membrane potential in 2nd chromosome lines of GtACR fluctuate similar to behavior?

In      Figure 5, Authors used a few light intensities to see the effect, please mention light intensities in the graphs, instead of labeling dim, low,      medium and bright. Surprisingly, heart rate cessation is occurring only at the brightest light intensity in case of GtACRs, suggesting that these constructs are not as potent as they are in the nervous system.

In section 3.3, authors measured membrane potential in both cardiac and body wall muscles, for readers it will be grateful to have some pictures of real prep.

In the line from 336-338, authors mentioned showing percent change in the membrane potential and point out Figure 6B,  but such graph where % change in membrane potential is plotted is missing.

In line 362, please mention the panel next to figure 8, perhaps 8A.

In Line 380, please re-write. You      probably meant that heart beat rapidly       in normal saline

Discussion:

The discussion is informative but general, the authors have not discussed why lines inserted on 2nd and 3rd chromosomes having different effects on heart rate, the shortcomings in the study and future plans.

Reviewer 2 Report

The presented manuscript discusses optogenetic manipulation of cardiac and body wall muscles in Drosophila. To this end, the authors activated light-gated anion channels (GtACR1 and GtACR2) and Halorhodopsin in the tissues of interest, and studied membrane potential changes and muscular contractions. They observed membrane depolarization upon low-intensity light activation of anion channels, in line with the reversal potential for chloride in myocytes. Cardiac contractions were inhibited both by high-intensity illumination of GtACR1-expressing cells, and by activation of Halorhodopsin leading to membrane hyperpolarization.

Major concerns:

1)      The literature being cited is inappropriate in a number of places, and important publications are not considered. The key findings of this manuscript have previously been described for other cell types or other species, and these papers need to be included and discussed. Additionally, there is a bias towards citing own papers. For statements without references, it is not clear if these are speculations or what they are based on.  

2)      There seem to be effects of GtACR1 and GtACR2 expression on cardiac muscle even before light application (section 3.3.2). If these effects are based on background illumination required for dissection, the experiments would need to be repeated while performing the experiments under red light (> 600 nm). Also the membrane potentials reported in this section (3.3.2) are quite depolarized, both for the ACR and Halorhodopsin experiments. Are these real values or is there a systematic error of measurement? The positive values are especially difficult to understand in the cases where the tissue is contracting, because the pacemaker channels would most likely be inactivated at such positive membrane voltages, which would prevent beating.

3)      If you observe depolarized currents using GtACR1/2, did you try to elicit action potentials/ pace the cells? This would be the natural control experiment to be performed.

Detailed review:

Abstract:

“Unlike body wall muscles in most animals, the equilibrium potential for Cl- is more positive as compared to the resting membrane potential in larval Drosophila” – Do you have any prove for this statement (values/ references from other species)? Otherwise, please delete the comparison.

“Cl- counteracting the influx of Ca2+” – I am not sure if this is the correct explanation. I would assume that the Ca2+ channel might be inactivated. The neurophysiologists often use the term “shunting the membrane potential to the reversal potential for chloride”, this might be useful to explain the drop in input resistance.

Introduction:

-        The clinical introduction is quite remote from the topic of the actual work, which describes Drosophila (a not clinically relevant model system).This part would benefit from proof-reading by a cardiologist or clinically aware scientist.

-        “A large percentage of direct cardiac disorders involve issues in electrical pacing of the heart.” Please rephrase. Which disorders are you referring to? When talking about cardiac pacemaking, please do not use the term electrical “shock”. You may want to use it for “defibrillation” instead?

-        “pharmacological treatments for these issues”. Please be more concrete or delete.

-        “Furthermore, inhibition of electrical activity is not possible with electrical shocking of the tissue.” But this is exactly what is done when defibrillating. Maybe “prolonged inhibition”? What would this be useful for?

-        References 1, 10: The first two cardiac optogenetics papers were Bruegmann et al. Nature methods 2010 and Arrenberg Science 2010. Reference 10 is inappropriate here.

-        “pacing cells” – pacemaker cells? Or excitable cells?

-        Lines 54 – 70: Please decide if citing literature for optogenetics in general or in Drosophila (or both). Please cite literature more systematically. The first two Channelrhodopsin papers were published by Nagel et al. (Science 2002, PNAS 2003) with subsequent use in neurons (e.g. Boyden, Deisseroth 2005). The first publications of anion channels (ACR) were Wietek Science 2014, Berndt Science 2014, and reference 12. The first use of ACR in cardiac muscle is Govorunova Sci. Rep. 2016 (6, 1–7).

-        “As the excitation wavelength is different”- how different?

-        “extremely fast” – please provide values for on- and off-kinetics

-        References 17/18: please replace by appropriate reference for the pumbing function of HR, e.g. Schobert and Lanyi, JBC 1982 (257: 10306)

-        References 27-37. Only own work cited. Please reduce the number of papers to the necessary (max. 3-4).

-        References 40, 41: Inappropriate here. One of the first work using ChR in Drosophila was Schroll et al Current biology 2006 (16: 1741-7)

-        “This can be explained by a chloride equilibrium potential above resting membrane potential in heart muscles, in contrast to neurons.” – There have been a number of papers describing depolarizing effects of anion channelrhodopsins in neurons. Please discuss these papers in either the introduction or discussion.

e.g.:

Mahn, M., Gibor, L., Patil, P., Cohen-Kashi Malina, K., Oring, S., Printz,

Y., et al. (2018). High-efficiency optogenetic silencing with soma-targeted

anion-conducting channelrhodopsins. Nat. Commun. 9:4125.

doi: 10.1038/s41467-018-06511-8

Mahn, M., Prigge, M., Ron, S., Levy, R., and Yizhar, O., (2016). Biophysical

constraints of optogenetic inhibition at presynaptic terminals. Nat. Neurosci.

19, 554–556. doi: 10.1038/nn.4266

Malyshev, A. Y., Roshchin, M. V., Smirnova, G. R., Dolgikh, D. A., Balaban,

P. M., and Ostrovsky, M. A., (2017). Chloride conducting light activated

channel GtACR2 can produce both cessation of firing and generation of action

potentials in cortical neurons in response to light. Neurosci. Lett. 640, 76–80.

doi: 10.1016/j.neulet.2017.01.026

Messier, J. E., Chen, H., Cai, Z.-L., and Xue, M., (2018). Targeting light-gated

chloride channels to neuronal somatodendritic domain reduces their excitatory

effect in the axon. Elife 7:e38506. doi: 10.7554/eLife.38506

Wiegert, J. S., Mahn, M., Prigge, M., Printz, Y., and Yizhar, O.,

(2017). Silencing neurons: tools, applications, and experimental

constraints. Neuron 95, 504–529. doi: 10.1016/j.neuron.2017.

06.050

-        “Such procedures to dampen the heart rate may be possible to implement in mammals as well as humans in the future.” Please rephrase. Please take into account that such experiments have already be done in mammals. Please have a special look into the following paper describing ACR-mediated depolarization in cardiac muscle: Kopton et al. Cardiac Electrophysiological Effects of Light-Activated Chloride Channels. Front Physiol. 2018 Dec 17;9:1806

Methods:

-        Do you use cell-type specific expression? This is not evident for non-Drosophila specialists. You may want to use “eNpHR” instead of “Halorhodopsin” throughout the manuscript.

-        200 µM all-trans retinal is a very high concentration (1µM should be sufficient)

-        Please include space between numbers and “nm”

-        Why do you provide light intensities in µM? Light intensities seem quite low in general, especially for NpHR

-        Did you use a Ca2+ buffer for the Ca2+-free solutions?

-        “cardiomodulators” –please find better term

-        “Salts for the saline and serotonin were obtained from Sigma Chemical Company.” Please just provide in brackets.

-        “Measures” – replace by “measurements or recordings” (also in other sections)

-        megaOhm – use MOhm

-        “was shined” – was applied

-        “each preparation had two trials” – consisted of / was exposed to…

-        Statistical analysis: Did you always use Sing text? In this case pleas delete the two other tests. Please indicate statistical significance in all figures showing paired datasets.

Results:

-        Please explain the difference between the two insertion sites for the transgenes. Did you quantify expression? This would be very useful to understand the differences.

-        “One of the 11 larvae in the GtACR1 attP40 line was able to continue beating during green light exposure but at a lower rate.” This is a contradiction to the statement above (“All…”). The entire first paragraph of 3.1 would greatly benefit from further structure.

-        3.2: Why were non-responders excluded from the datasets? These must be included! Please include in Figure 3.

-        “Crosses using only virgin females of non-curly in both lines, GtACR1ATTP40 and GtACR2ATTP40, and non-stubble 24B males produced larvae sensitive to light. “ Please explain this finding (could also be done in the discussion)? Did you expect this?

-        Please try to stream-line the entire section to minimize repetition.

-        Reference 49: please delete.

-        3.3.1: “small spontaneous quantal responses of vesicle fusion” – this is already an interpretation. Please describe observations. (postsynaptic currents?)

-        “short-circuiting the additional Na+-induced depolarization attempted by the opening of the ionotropic glutamate receptors” – Na+ channels are probably inactivated at depolarized potentials. Please find alternative term for “short-circuiting”.

-        “Thus, it would appear the GTACR2-chloride channels were open to decrease input resistance and the expected depolarizing synaptic quantal responses.” – This is not clear to the reviewer.

-        Please quantify the amplitude and frequency of the quantal events before and during light application. It is stated: “The quantal response increase in amplitude”. This needs quantification and statistical testing. It would then be quite interesting.

-        Figure 6: please state light intensity.

-        Figure 7A: please rephrase A)

-        3.3.2: “higher intensity of white light exposure needed for dissection”. Could the activity be restore by keeping the tissue in the dark? If not, these experiments should be repeated by dissecting the tissue under red light. Both GtACR1 and GtACR2 are extremely light sensitive.

-        “reduced movement due to dissection pins”. Is this true? Also, why is this stated here? The entire section needs more stream-lining/ shortening. Please remove unnecessary information.

-        Why are the membrane potentials so depolarized in these measurements?

-        Figure 9 A2) “sustained hyperpolarization”. Please reconsider since hyperpolarization is < 1 mV. This is a very small effect. Is there potentially a systematic error with the membrane voltage scaling?

Discussion

-        General comment: would benefit from shortening. Please only discuss relevant information in respect to the current study. Also, please include proof/ references/ support for all statements.

-        “A similar mechanism of preventing depolarization-induced contractions would likely occur in mammals”. Please discuss and compare to Kopton et al. Front Physiol. 2018 Dec 17;9:1806.

-        “The likely higher expression of the proteins and larger surface area of the body wall muscles accounts for the …” This is not clear at all. How would the larger surface area affect the membrane potential. Is there prove for higher expression levels?

-        Again “hyperpolarization in neurons” – see above.

-        “effective means of controlling the heart even under the conditions which accelerated the rate”. This is based on the underlying biophysical principles. Why should this be different at different rates.

-        “overriding intracellular cascades” – not correct.

-        Targeting to other cellular membranes – there is a lot of literature about this, especially for NpHR, please refer to it. Otherwise this part might not be necessary.

-        “secondary consequences” – not relevant for this paper

Final comment: Please ensure proof-reading of the manuscript by all co-authors before resubmission.

Reviewer 3 Report

Stanley et al. investigate the effects of light-activated anion conducting Channelrhodopsins GtACR1 & 2 and the chloride pump halorhodopsin on the heart rate and on heart muscle as well as body muscle physiology in Drosophila larvae. They find that activating GtACRs leads to depolarization of heart muscle cells, due to a depolarized reversal potential of chloride in these cells. At low ionic flux (e.g. under low-light illumination) this leads to tachycardia in dissected larvae, while high ionic flux leads to a complete cardiac arrest – most likely due to strong shunting effect. In contrast, halorhodopsin was hyperpolarizing under all intensities as expected from a chloride pump leading to cardiac arrest under intense illumination.

The findings related to the seemingly paradoxical behavior of the GtACRs are interesting, supporting earlier studies that made similar observations. Therefore, this study is potentially interesting for publication in Methods and Protocols.

However, this also brings me to my first point of criticism. The authors largely ignore already existing literature that used optogenetic inhibitory tools to modulate heart function in various model systems (Arrenberg et al., 2010; Govorunova et al., 2016; Kopton et al., 2018) and more importantly, they missed a previous study that already showed effects similar to their findings with GtACRs in mammalian hearts (Kopton et al., 2018). The authors need to provide a better background on already existing knowledge in the field and appropriately set their work in this context.

I also tend to disagree with the statement that channels, “that reduce activity have not been extensively explored so far”. There are many studies exploring the application of light-activated anion conducting Channelrhodopsins for neuronal silencing and its limitations (Berndt et al., 2015; Mahn et al., 2018; Mahn et al., 2016; Malyshev et al., 2017; Messier et al., 2018), and silencing with light-activated anion channels has been explored in Drosophila (Mohammad et al., 2017; Wietek et al., 2017). In general, the citations regarding neuronal silencing appear to be somewhat randomly selected. For example, why do the authors cite Banghart et al. 2004 (ref. no. 14)? This one has nothing to do with “chloride conducting Channelrhodopsins” or “halorhodopsin”, which they discuss in the preceding sentence. Another example is the citation of the review article by Owald et al. in line 82 (ref. no. 42). It does not say anything about heart rate modulation with ChR2-XXL and apart from mentioning it once, does not give any information on this tool. Also, why do the authors mention iC1C2 in line 52? Is this a randomly selected example representing all the engineered anion conducting Channelrhodopsins?

My second and major concern is related to the way the optogenetic manipulation experiments in larvae are done. First, the authors used white light to illuminate the animals in order to be able to image them. Although they use low intensities, this might still have an effect on GtACR-expressing hearts. Due to their extremely high conductance (and thus light sensitivity), GtACRs already affect neuronal activity under continuous illumination at intensities below 1 uW/mm^2 (Govorunova et al., 2015), close to the with light intensity used here. The low initial heart rate of dissected larvae expressing GtACRs (Fig. 3), but not NpHR (Fig. 4) under initial low-level white light might be explained by this. Second, because no optical filters are used in front of the camera, the stimulation light directly affects the camera image (and sometimes completely saturates it), as seen in the supplementary videos. This makes accurate analysis of the heart beat rate impossible. Finally, light delivery appears not to be very precisely controlled in space and time (do the authors deliver the light manually?). In fact, there is no description of the imaging procedure at all in the Methods section of the paper. Also, there is no description how exactly these experiments were analyzed. The proper way to do such an experiment is to illuminate the larvae with infrared light and to block any stimulation light from entering the camera with a long-pass optical filter. See here, for example (Mohammad et al., 2017). This allows continuous imaging without activating the optogenetic actuators and without interference from the stimulation light. The authors need to perform at least the experiments presented in Fig. 3 again under appropriate conditions and discuss the potential methodological limitations of the original experiments in the manuscripts.

A third concern relates to the selection of data included in the analysis. In lines 234-236 the authors state that preparations initially not responding to light were excluded. What is the reason for this? Was this because the animals were not expressing GtACRs or was expression toxic?

Finally, the authors describe different effects of GtACR activation and NpHR activation on quantal responses in their recordings from the body wall muscles. These phenomena are interesting and further corroborate the authors hypothesis that GtACRs can be inhibitory despite their depolarizing action. Thus, these measurements should be quantified and plotted in addition to the effects on the membrane potential.

Minor issues:

Figures 2-4 should be merged into one figure, same for figs. 6 & 7 and 8 & 9.

A two-tailed t-test is not appropriate for the data shown in fig. 5, since more than two conditions are compared.

Figure labeling mistakes:

-          y-axis figs. 6-9: “potental” should be “potential”

-          x-axis figs. 6-9: change to “dark” and “blue light”

References

Arrenberg, A.B., Stainier, D.Y., Baier, H., and Huisken, J. (2010). Optogenetic control of cardiac function. Science 330, 971-974.

Berndt, A., Lee, S.Y., Wietek, J., Ramakrishnan, C., Steinberg, E.E., Rashid, A.J., Kim, H., Park, S., Santoro, A., Frankland, P.W., et al. (2015). Structural foundations of optogenetics: Determinants of channelrhodopsin ion selectivity. Proc Natl Acad Sci U S A.

Govorunova, E.G., Cunha, S.R., Sineshchekov, O.A., and Spudich, J.L. (2016). Anion channelrhodopsins for inhibitory cardiac optogenetics. Scientific reports 6, 33530.

Govorunova, E.G., Sineshchekov, O.A., Janz, R., Liu, X., and Spudich, J.L. (2015). NEUROSCIENCE. Natural light-gated anion channels: A family of microbial rhodopsins for advanced optogenetics. Science 349, 647-650.

Kopton, R.A., Baillie, J.S., Rafferty, S.A., Moss, R., Zgierski-Johnston, C.M., Prykhozhij, S.V., Stoyek, M.R., Smith, F.M., Kohl, P., Quinn, T.A., et al. (2018). Cardiac Electrophysiological Effects of Light-Activated Chloride Channels. Front Physiol 9, 1806.

Mahn, M., Gibor, L., Patil, P., Cohen-Kashi Malina, K., Oring, S., Printz, Y., Levy, R., Lampl, I., and Yizhar, O. (2018). High-efficiency optogenetic silencing with soma-targeted anion-conducting channelrhodopsins. Nature communications 9, 4125.

Mahn, M., Prigge, M., Ron, S., Levy, R., and Yizhar, O. (2016). Biophysical constraints of optogenetic inhibition at presynaptic terminals. Nat Neurosci 19, 554-556.

Malyshev, A.Y., Roshchin, M.V., Smirnova, G.R., Dolgikh, D.A., Balaban, P.M., and Ostrovsky, M.A. (2017). Chloride conducting light activated channel GtACR2 can produce both cessation of firing and generation of action potentials in cortical neurons in response to light. Neurosci Lett 640, 76-80.

Messier, J.E., Chen, H., Cai, Z.L., and Xue, M. (2018). Targeting light-gated chloride channels to neuronal somatodendritic domain reduces their excitatory effect in the axon. eLife 7.

Mohammad, F., Stewart, J.C., Ott, S., Chlebikova, K., Chua, J.Y., Koh, T.W., Ho, J., and Claridge-Chang, A. (2017). Optogenetic inhibition of behavior with anion channelrhodopsins. Nat Methods 14, 271-274.

Wietek, J., Rodriguez-Rozada, S., Tutas, J., Tenedini, F., Grimm, C., Oertner, T.G., Soba, P., Hegemann, P., and Wiegert, J.S. (2017). Anion-conducting channelrhodopsins with tuned spectra and modified kinetics engineered for optogenetic manipulation of behavior. Scientific reports 7, 14957.

Round 2

Reviewer 1 Report

Response to revision:

Title: Thanks for accepting and changing the title

Introduction:

Introduction is much better now with added information and also well cited. Few minor changes:

English is still a problem:  I am citing few examples here, I will suggest authors to have input from colleague who is a native speaker or editing service. I am underscoring the correction.

Line 67-68: --should be shown to work, ...within a fraction

2. Citations: At some places order of citations is not correct: 

Line 65, citations should be 25, 31

Methods:

Method section has improved,  There are many typos and grammar issues: To name a few:

Line 141: Double sided sticky tape

Line 142:  ..ventral side of the larva on the tape

Line 145: ...while the Heart Rate (HR)

Line 147:.. This  was procedure was repeated..

Line 163: ..the saline mentioned above….

Line 168-170: The sentence ..”these steps were repeated until ten…….” has been repeated twice, once at line number 147-148 and and at line 168-170. Please remove it from one place.

Results:

Line: 195, double sided sticky tape

Line 202: ...behavior was seen observed in two of the…

Line 231-232: Please change ..”This has been noted for Canton-S in the past” with “ This has been shown previously.

Line 347: The eNpHR line had has been investigated

Line 415: ---instead of pick back up, you may say “start beating again”

Figures:

Figure-1: Legend: Ventral dissection representation of a third instar larvae.  Remove procedure from the legend and move it to the method section.

Figure 2: Legend: Heart rate of the intact larvae from gene line genotype before and after---

In Panels A1-E2,  Please indicate genotypes on top of each panel, eg. 24B x GtACR2, or w1118 x 24B etc

Figure 2-3, similar to figure 4, Please indicate the intensity value of light of both blue and green lights either on figure panel itself or in the legend, the author has already mentioned it in the method section, it will be handy for the reader to know what are the look at.

Figure 4: Please indicate full genotype on top of the panel A-C,

Figure 5: Please indicate the Rp value for the panel B for the second change in potential. For Panel C1-C2, please write full genotype on top of the panel as also mentioned earlier.

Figure6-7: Consistent with Figure 5, please indicate Rp and change in potential in Panel A.

Analysis: I will appreciate if authors can provide Mean + SEM or Mean + CI in all the graphs.

Reviewer 2 Report

Major points:

The introduction is still poorly written and is far from the topic of the article. Most importantly, one needs to distinguish between pacemaker disturbances and atrial and ventricular arrhythmias. In analogy it would be logic to distinguish between atrial and ventricular pacemakers and ventricular defibrillators. The new part on pharmacological treatment of hypertension does not fit to the presented topic and is misleading.  Please carefully revise the entire introduction using proper scientific terminology.

The new paragraph on the use of all-trans retinal is very misleading. Retinal is essential for light sensitivity of microbial opsins. Please correct the respective incorrect statements.

The presented measurements of cardiac muscles are not clear. Why are the membrane potentials so depolarized already in the dark? Are there differences in resting membrane potentials between the different lines? Why are there differences in beating (before light)? Why are light-induced effects on membrane potential so small, especially for the NpHR lines? How can such small effects inhibit activity? What is the effect on Ca2+ on cardiac action potentials in Drosophila? Please carefully revise the respective parts in the results and discussion section.

Please improve language and style of the entire manuscript and use quantifications wherever possible.

Detailed comments:

Abstract:

-ion selective conductance – please rephrase. Many optogenetic acturators are not selective for a specific ion, e.g. ChRs are cation non-selective, Halorhodopsin are anion non-selective.

Introduction:

- “pacing cells in the desired direction” – cells are paced by initiating action potentials, which are always depolarizing.

-“within a fraction of seconds” – please be more specific; GtACRs work on the millisecond timescale.

- “as comparable indices to mammalian hearts” – please rephrase

- “investigations into in vivo studies” – please rephrase

-“which alter pacing of the cells and intact heart” – please improve sentence structure

Methods:

- please use space between number and units throughout

-        Please remove repetitions of words e.g. “The” line 127.

-        “The intensities of the LEDs are similar to the strengths used in previous studies in our laboratory [47,52,57-60].” Please remove this sentence and reduce number of references of own work.

-        “The light was moved to a distance which produced a focused beam directly on the heart tube [61].” – Please quantify distance and light intensity in the object plane.

-        “Sigma Chemical Company” – Please use correct name of companies.

-        Line 174: Please correct name of amplifier.

Results:

-        Figure 2 and 3: Please streamline the text of the figure legends, especially for the controls. Please avoid repetitions.

-        “ventral aspect of larvae” – please rephrase.

-        “the values were compared to the maximum response during a 10 sec light exposure.” – please specify which values?

-        “A representative recording in the body wall muscle” – what is measured? Membrane voltage? Please improve terminology.

-        Rapid depolarization – please determine and state time constants of de- and repolarizations

-        “dark exposure” – please rephrase.

-        “The focus of this procedure was to show that activating the GtACR1 and GtACR2 channels with light results in a depolarization of the body wall muscles in the majority of cases.” How can you know that the effect is depolarizing  before the actual experiments. Please rephrase as you performed the experiments to test an hypothesis.

-        Figure 5: Please improve Figure legend,  second sentence.

-        “During the light-induced hyperpolarization state there is an increased driving gradient for the Na+ influx through the glutamate receptors.” This is not evident from the example trace. Have you quantified quantal release during and between illumination? Please only describe observed phenomena in the results section.

-        Why is the resting membrane potential > -30 mV in cardiac muscle? Are these normal values for cardiac muscle in Drosophila? If so, Drosophila is probably an inappropriate model to study cardiac electrophysiology.

-        Figure 8: Are there systematic differences between the resting membrane potential of the GtACR lines and the halorhodopsin lines already in the dark? If so, please quantify. In A2, the observed hyperpolarization is very small, if not neglectable (~0.5 mV). Please say so. How can this prevent action potentials? This is not clear at all. Please also discuss these points in the discussion.

-        When lowering Ca2+ in the extracellular solution, why do you inhibit action potentials? (see line 414). Are those not driven by the fast Na channel? Inhibition of contractions might occur while retaining electrical activity. Why is this not always the case?

Discussion:

-        “when the additional Na+-induced depolarization occurred by the opening of the ionotropic glutamate receptors” – this is not clear to reviewer. Please explain better.

-        “quantal responses would be smaller” – but were there smaller? Did you quantify and tested this statistically?

-        Lines 486-490. This statement is not clear. Please revise. What were the differences in resting membrane potential? Doesn’t the conductance depend on the presence of Chloride channels? How well can this be transferred to the Drosophila model?

-        “hyperpolarization is induced with a Cl- pump [49,52]. Consequently, activation of GtACR1 and GtACR2” – Please improve logic of paragraph.

-     

Author Response

See attached PDF

Reviewer 3 Report

I still have some issues with the statistics. A sign pairwise test cannot be applied to experiments with more than 2 conditions as presented in figures 3 and 4. 

Moreover, the quantal events should still be quantified, even if they are going to 0 with GtACR. Just showing example traces and making general statements is not a very scientific approach. For example, it is not clear from theses examples, if NpHR still has effects due the membrane hyperpolarization.

Author Response

see attached PDF
